# Snorkeling Strategy: Tolerance to Flooding in Rice and Potential Application for Weed Management

**DOI:** 10.3390/genes11090975

**Published:** 2020-08-22

**Authors:** Tiago Edu Kaspary, Nilda Roma-Burgos, Aldo Merotto

**Affiliations:** 1Instituto Nacional de Investigación Agropecuaria, INIA, La Estanzuela, Colonia 70006, Uruguay; tkaspary@inia.org.uy; 2Department of Crop, Soil, and Environmental Sciences, University of Arkansas, Fayetteville, AR 72701, USA; nburgos@uark.edu; 3Department of Crop Sciences, Agricultural School, Federal University of Rio Grande do Sul, Porto Alegre 90040-060, Brazil

**Keywords:** flooding tolerance, hypoxia, rice diversity, SUB1, weedy rice

## Abstract

Flooding is an important strategy for weed control in paddy rice fields. However, terrestrial weeds had evolved mechanisms of tolerance to flooding, resulting in new ‘snorkeling’ ecotypes. The aim of this review is to discuss the mechanisms of flooding tolerance in cultivated and weedy rice at different plant stages and the putative utility of this trait for weed management. Knowledge about flooding tolerance is derived primarily from crop models, mainly rice. The rice model informs us about the possible flooding tolerance mechanisms in weedy rice, *Echinochloa* species, and other weeds. During germination, the gene related to carbohydrate mobilization and energy intake (*RAmy3D*), and genes involved in metabolism maintenance under anoxia (*ADH, PDC*, and *OsB12D1*) are the most important for flooding tolerance. Flooding tolerance during emergence involved responses promoted by ethylene and induction of *RAmy3D*, *ADH*, *PDC*, and *OsB12D1*. Plant species tolerant to complete submersion also employ escape strategies or the ability to become quiescent during the submergence period. In weedy rice, the expression of *PDC1*, *SUS3*, and *SUB1* genes is not directly related to flooding tolerance, contrary to what was learned in cultivated rice. Mitigation of flooding tolerance in weeds could be achieved with biotechnological approaches and genetic manipulation of flood tolerance genes through RNAi and transposons, providing a potential new tool for weed management.

## 1. Introduction

Partial or complete flooding hampers fundamental plant processes, such as respiration and photosynthesis, because of low oxygen (O_2_) diffusion in water. In addition to O_2_ deficiency, flooding promotes the accumulation of CO_2_, methane, ethylene, hydrogen sulfide, and hydrogen, reducing aerobic respiration [1]. Some plant species, mainly rice, have been selected under conditions of hypoxia (oxygen deficiency) or anoxia (absence of oxygen) [2]. Evolution under oxygen-deficient environments occurs by gradual adaptation to low-energy status as a consequence of flooding, or due to complex processes related to the lack of O_2_ and mitochondrial respiration, stimulating mechanisms to generate ATP through anaerobic pathways and mechanisms of quiescence or escape from flooding [3]. Flooding induces complex responses that vary depending on the severity and duration of stress, plant growth status, as well as specie and cultivar characteristics. Several *Oryza* species are naturally tolerant to flooding. This characteristic allowed the domestication of rice in floodplains, or seasonally inundated areas and lowland areas across the globe.

Extensive studies on the physiological effects of flooding stress on soybean and wheat have been conducted and are continually pursued with the goal of developing flooding-tolerant varieties [4]. In addition, the flood-tolerant, non-crop species, such as *Echinochloa* spp. and *Cyperus* spp. (major weeds in rice), have also been investigated to gain an understanding of genes and genetic mechanisms for flood tolerance [5,6,7]. However, the majority of knowledge about flooding tolerance came from cultivated rice. A milestone achievement in this pursuit is the development of Swarna-Sub1 rice (Table 1). This cultivar tolerates complete submersion for two weeks and resumes regular growth afterwards [8,9]. Submersion-tolerant rice, with high yield, has long been sought for areas subject to deep flooding. Good yield of deep-water rice varieties in East India, for example, is between 2 and 3 t ha^−1^ [10]. Flooding tolerance is also present in several weed species, such as *Echinochloa* spp., *Sagittaria* spp., *Heteranthera sessilis, Leptochloa chinensis*, *Cyperus* spp., *Potamogeton distinctus*, *Ludwigia* spp., *Ipomoea aquatica*, and *Fimbristylis miliaceae* [6]. These weeds have submersion tolerance mechanisms, which could be called snorkeling strategy, which are strategies that guarantee its survival and perpetuation in flooded environments.

Flooding is the most efficient cultural method for controlling flood-sensitive weeds soon after rice sowing. Water management characterizes the different rice establishment systems, such as dry-seeding, water-seeding or transplanting [11]. These systems were developed to provide water for the rice crop and to control weeds. Therefore, although rarely acknowledged, water is a natural herbicide used in flooded crop agriculture. However, since rice domestication, weeds had co-evolved with rice, and flood-tolerant weeds became one of the main problems in rice production. Transplanting and water-seeding systems are more efficient for weed control than dry-seeding, especially for weedy rice, which is one of the most important constraints of rice production worldwide. However, weedy rice is also adapting to flooding [12,13], perhaps more than the other rice weeds, and this presents a serious challenge for the utilization of water management strategies to establish rice. The changes in metabolism and morphological growth processes of weedy rice that escape flooding suggests evolution for tolerance to this abiotic stress [5].

Current advances in biotechnology present the possibility of using molecular tools to understand flooding tolerance, and manipulate DNA and RNA for the development of modern snorkeling weed control methods [14,15]. One of these tools is RNA interference, which affects the silencing of a specific target trait, such as flooding tolerance. The use of interfering RNA (RNAi) and transposable DNA elements have been proposed and explored to mitigate the evolution of herbicide resistance in weeds [14]. Biotransformation of rice is saddled with the issue of gene flow to its weedy relative, which is the same species (*Oryza sativa* L.) or even to its wild ancestor (wild rice races) in regions of Asia where rice cultivation originated. Gene flow mitigation is a major biotechnological challenge and has been a high-impact research topic in the rice-weedy rice continuum for many years. One approach is to utilize RNAi technology to mitigate gene flow. An RNAi system composed of an unfitness gene together with another gene of interest, such as herbicide resistance, would reduce the spread of the transgene into weedy relatives. The gene silencing process has been pursued as an advanced biotechnological tool to reduce the gene flow consequences of transgenic crops [16], such as rice to weedy rice, sugar beet to wild beet, sorghum to johnsongrass, and canola to wild mustard [17,18]. In this case, the transgene silencer is inactive in the crop and is activated only in the weedy relative based on recognition of genetic differences in these plants. Artificial microRNA and anti-sense RNA techniques were used to silence the gene *SH4*, reducing seed shattering in weedy rice that hybridizes with rice crop [19]. The seeds of these plants will remain in the panicle and will be harvest together with the rice cultivated seeds decreasing soil seed bank and spread od weedy rice. The same approach can be used to mitigate the invasiveness of weedy rice in flooded environments.

The objectives of this review are to digest the current knowledge about the main mechanisms of flooding tolerance in cultivated and weedy rice at different plant stages and to present its putative utility for weed control.

## 2. Rice Flooded Strategy

Water deficit is the main constraint to food production. Rice, which is cultivated in flooded environments, generally does not have this limitation as of today for as long as there is ample irrigation water. However, flooding is common in low-lying or coastal areas and river floodplains during the rainy season. Seasonal floods cause partial or total submergence of the rice in early development stages, requiring extreme metabolic adaptations. Some rice populations have been identified with a natural capability to tolerate flooding [8,20,21]. 

Flood-tolerant rice can survive flooding during germination, early growth, or at the vegetative stage (Table 1) by fast mobilization of reserves to elongate their internodes under the flood, extending their shoots above the water to facilitate uptake of oxygen from the atmosphere and transport it to the roots via the aerenchyma [5,6,22]. This flood adaptation is described as low O_2_ escape syndrome (LOES), induced by ethylene accumulation and regulated by *SNORKEL* (SK) locus which induces the production of gibberellic acid and auxin [20]. These products collectively stimulate underwater growth and stretching [20,22]. Other rice varieties that can survive total flooding during vegetative stage have a tolerance strategy based on low O_2_ quiescence syndrome (LOQS) [20]. This tolerance mechanism is driven by ethylene, with the *SUB1* locus encoding three *Ethylene Response Fa*c*tor* (ERF-VIIs) genes, including *SUB1A*, which regulates metabolism and extends the survival period of plants under complete submersion up to two weeks. Growth and other energy-demanding processes are suppressed, allowing the conservation of resources until the flooding ends [23,24]. 

The development of rice genotypes with escape or quiescence strategy was done by selecting native varieties or by introgression of specific tolerance genes [8,25]. The cultivated and weedy rice genotypes and the putative mechanisms of flooding tolerance and the respectively associated genes are shown in Table 1. For example, rice varieties with escape strategy are related to IRRI 119 and IRRI 154 varieties, which are tolerant to flooding up to 30–50 cm. On the other hand, Baisbish and Rayada 16-3 varieties survive flooding over 50 cm [20,25]. However, the varieties with quiescence strategy, such as Swarna-Sub1 and IR64-Sub1 (Table 1) provide security in rice production in more than 20 million ha subjected to flash floods and deep flooding, specifically in Asia [25].

**Table 1 genes-11-00975-t001:** Rice genotypes identified with flooding tolerance in the different development stage. UFRGS, June 2020.

Timing of Flooding	Tolerant Genotypes	Main Traits Associated with Tolerance	Main Genes *	References
**Germination and early growth**	KHAO HLAN ON KHAIYAN F291 F274-2A 8391 8753 ITJ03 AV04	Early development of root and coleoptile underwater; Green shoots appear earlier; Rapidly growing coleoptiles	*RAmy3D *** *ADH1* *ADH2* *SUS1* *PDC* *SNORKEL1* *SNORKEL2* *OsTPP7*	[3,5,6,13,20,25,26,27,28]
**Vegetative**	FR13A KURKARUPPAN GODA HEENATI THAVALU IR64-SUB1 SWARNA-SUB1 SAVITRI-SUB1 CIHERANG	Reduced elongation; Slow carbohydrate consumption during submergence; Underwater photosynthesis; Chlorophyll retention underwater; Fast recovery	*SUB1A-1*	[8,9,21,23,24,29,30,31,32]
	IRRI119 IRRI154 JALMAGNA BAISBISH RAYADA 16-3 SUDU GRIES TIL BAJAL FULKARI TUNG LU 3	Fast internode elongation with rising water; Sufficient leaf area above water; Photosynthesis underwater; Kneeling ability when the water recedes; Large fertile panicles;	*SNORKEL1* *SNORKEL2* *ADH1* *ADH2* *SUS1*	[20,33,34,35,36]

* One or more genes can be associated with the same tolerance mechanism. ** RAmy3D (Rice α-amylases); ADH1 (alcohol dehydrogenase1); ADH2 (alcohol dehydrogenase2); SUS1 (sucrose synthase 1); PDC (pyruvate decarboxylase); SK1 (Snorkel1) SK2 (Snorkel2); OsTPP7 (trehalose-6-phosphate phosphatase); SUB1A (Submergence-1).

In flooded rice culture, the primary biological constraint is competition with flood-tolerant weeds. In the rice paddy, yield losses due to weeds vary from 20% to 60% for transplanted rice and from 30 to 100% for dry-seed rice [11]. In flooded rice, correct management of water depth is the most efficient tool for the management of non-aquatic weeds. Flooding is effective in controlling major weeds, such as *Echinochloa colona*, *E. crus-galli*, *Fimbristylis miliacea*, *Cyperus iria*, and weedy rice, and has been used for this purpose since rice domestication [7,37]. Flooding during the germination and emergence stages is more effective in controlling *E. crus-galli*, *Cyperus difformis*, *F. miliacea*, and weedy rice than late flooding [37]. Continuous exposure to flooding resulted in the evolution of flood tolerance in some weeds, making the control of these species less effective. Fallow periods due to water limitations or farm structure can reduce selection pressure, reset the weed community, and delay the evolution of flooding tolerance in weeds. However, the current intensification of agriculture to obtain higher grain yields per unit area entails continuous rice cultivation in the same land, resulting in increasing selection for flooding tolerance. 

The evolution of *C. rotundus* in rice fields of the Philippines illustrates the extreme metabolic and morphological modifications that weedy species undergo to adapt to extreme environments. *C. rotundus* is an upland weed, unlike its cousins *C. esculentus* and *C. iria*, which are naturally adapted to saturated or water-logged soil. The latter two are, historically, problem weeds in rice. After decades of planting rice in rotation with onions or other vegetables, *C. rotundus* became increasingly dominant in the rice rotation as more tubers were able to germinate and emerge under flood. It was then documented that this new flood-tolerant ecotype of *C. rotundus* is able to generate energy for growth and development through anaerobic metabolism [38]. Recently, weedy rice ecotypes were identified with the capacity to establish under submergence, although the mechanisms of tolerance are not known [12]. Note that the natural biology of weedy rice (and cultivated rice) is such that it can germinate under flood only if the seed is on the soil surface, but not when the seed is buried. Understanding the mechanism of flooding tolerance in rice weeds could inform the development of alternative management methods for these species.

## 3. Mechanisms of Tolerance to Flooding in Rice and Weedy Species

Coping mechanisms in flood-tolerant species occur singly, or in concert. In most situations, a series of stepwise mechanisms occurs at seed germination, seedling emergence, and initial plant growth. In rice, several primary genes involved in flooding adaptation have been identified and preliminary theories about how these genes affect flood tolerance have been proposed. A diagram summarizing the main genes involved in rice tolerance to flooding at different stages of development, and the potential advantages conferred to weeds, is presented in Figure 1.

### 3.1. Tolerance to Flooding during Germination

Seed germination is driven by a complex regulatory system. The energy demand during germination is high; numerous exogenous and endogenous factors affect this process, mainly associated with oxygen availability. Germinating seeds in normal oxygen conditions (O_2_ ≥21%) derive energy via the electron transport chain and oxidative phosphorylation using O_2_ as final acceptor of electrons. Under anaerobic condition, the electron transport chain and oxidative phosphorylation are inhibited by lack of O_2_; the plant then turns on glycolysis to generate energy. Glycolysis is an inefficient process of generating ATP; thus, anoxia prevents germination of most species. To obtain sufficient energy for normal physiological function in flooded conditions, more carbohydrates are mobilized toward glycolysis; anaerobic respiration, or fermentation, is activated. The evolved ability to maintain metabolic processes under low O_2_ allows the germination and emergence of some species in flooded conditions. Examples of these species are *Echinochloa spp.*, *Arundinella anomala*, *Sagittaria montevidensis*, *S. trifolia*, and weedy rice, which are major weeds of rice [12,27,39].

The evolved flood-tolerant ecotype of *C. rotundus* stores more non-structural carbohydrates, which serve as a substrate for anaerobic metabolism, compared to the normal flood-sensitive ecotype [38]. Tolerant ecotypes have larger tubers with about twice more non-structural carbohydrates than susceptible plants [38]. However, just producing more carbohydrates does not guarantee an increase in anaerobic metabolism. The plant also has to be able to convert these carbohydrates into glucose, which is the form of sugar that is transported and used in glycolysis. Therefore, adequacy in the complex network of metabolic processes is required for the evolution of tolerance to submersion. Low oxygen triggers the formation of ethylene, which induces the expression of genes related to carbohydrate conversion and maintenance of anaerobic respiration. Rice seeds that germinate well under low oxygen supply have higher expression of *RAmy3D* than in rice varieties intolerant to hypoxia [3]. Overexpression of *RAmy3D* gene in the coleoptile and endosperm of transgenic rice genotypes increased the germination capacity under anoxic conditions [16]. *RAmy3D* codes for alpha-amylase, which is involved in the catabolism of carbohydrates during germination. Concomitantly, the sucrose synthase (*SUS3*) gene promotes the production of soluble sugars when oxygen is deficient [23]; thus, providing the substrate for ATP production under oxygen-deficient environments (Figure 1).

The ATP supply for metabolism under anoxic conditions is enabled by the maintenance of the glycolytic pathway that promotes the regeneration of NAD^+^. This process is enabled by the consumption of pyruvate by fermentation, avoiding the accumulation of this by-product of glycolysis and decrease of the glycolytic flux due to NAD^+^ limitation [40]. Lactic acid fermentation starts with the production of lactate (facilitated by lactate dehydrogenase, LDH), which acidifies the cytoplasm and activates alcoholic fermentation by the action of pyruvate decarboxylase (PDC) and alcohol dehydrogenase (ADH), and producing ethanol [26]. These reactions are maintained through the oxidation of NADH in the mitochondrial matrix. In this process, NADH and oxaloacetate are reduced, and reverse Krebs cycle reaction is catalyzed by malate dehydrogenase, regenerating NAD+, thereby maintaining the production of ATP [40].

*Echinochloa crus-galli*; *Arundinella anomala* and cultivated rice tolerant to flooding have higher activity of *ADH*, aldehyde dehydrogenase (*ALDH*), and *PDC* than susceptible genotypes [3,6,27]. In a complementary or alternative process, the overexpression of *OsB12D1* allowed germination and emergence of rice under anoxic conditions. This gene encodes an enzyme related to the increase of electron transport in the mitochondria through Fe mediation, allowing ATP production even under anoxic conditions [41]. 

A recent study on 158 weedy rice ecotypes, as well as indica and japonica rice cultivars, revealed that flooding tolerance during germination and early growth is associated with high expression of genes evolved to mobilize reserves of *RAmy3D* and *OsTPP7*, and acquisition of anaerobic respiration via induction of *ADH1* and *ADH2* [13], as described above. The process culminates with the rapid elongation of coleoptile and rapid emergence enabled by the *SNRKL1* gene. The induction of genes *PDC1, SUS3*, and *SUB1* is not directly related to flooding tolerance in weedy rice [13], which is contrary to the importance of these genes flood tolerance in cultivated rice (Table 1) [26]. Therefore, flood tolerance in weedy rice is driven by a different set of genes than those in cultivated rice.

### 3.2. Tolerance to Flooding during Seedling Emergence

When germinating under flood, the rice coleoptile has to elongate to reach the water surface to perform aerobic oxidative phosphorylation. Therefore, the emergence requirements have to be supplied quickly, before the energy reserves in the seed or vegetative propagules are exhausted [5]. Coleoptile elongation, and the speed at which it occurs, constitutes one of the ‘snorkeling’ strategies for flooding tolerance. Seedling emergence in flooded conditions is facilitated by ethylene-dependent responses [23]. In one study under flooded soil, ethylene was not detected in seeds of rice cultivars IR42 and FR13A not adapted to flooding at germination, but was abundant after germination in seedlings of the flood-tolerant genotype KHO [26]. The delay in ethylene production suggests that the source of ethylene is probably the embryos that grow under oxygen deficiency, being more related to post-germination growth rather than germination *per se* [26].

Ethylene acts as a stimulus for the mobilization of seed reserves after germination based on the activation of alpha-amylase enzymes [23,25]. Similar to what occurs in germination, high expression of the *Ramy3D* gene is observed in the coleoptile and radicle of rice seedlings under anoxia (Figure 1). The expression of this gene was up to 592 times higher in the tolerant than in the susceptible genotype of rice at 8 d after seed imbibition [42]. Also associated with anoxia-tolerance in rice is the loci OssTPP7 (Gene *OsTPP7*), which encodes the enzyme trehalose-6-phosphate, responsible for increasing the conversion of starch to glucose [28]. This gene was identified in the flooding-tolerant cultivated rice Khao Hlan On, but was absent in the susceptible cultivar IR64, implicating OsTPP7 in flood tolerance [28]. The higher availability of sugar promoted the elongation of coleoptile, resulting in emergence and high tolerance to submersion [2,28]. Thus, these mechanisms enable seedling root and shoot growth of tolerant rice genotypes in flooded conditions. Tolerance to flooding during emergence of *Echinochloa* spp. Was linked to increased activity of LDH, ALDH, ADH, and PDC enzymes [6,23]. The growth of roots and shoots of flood-tolerant *E. crus-galli* was not affected by 100 mm depth of flood for 7 d after sowing [6]. 

Increased structural proteins and key enzymes have been related to continued coleoptile growth of rice seedlings under anoxia and may be the same factors involved in weed adaptation to flooding during emergence. The action of the tubulin *a-1* chain (*TUBA1*) and actin factor 4 (*ADF4*) genes on the rapid elongation of coleoptile under anaerobic conditions has been proposed because both are upregulated under anoxia during rice emergence [43]. Tubulins enable cell division and elongation, as well as ADF4, is involved in the regulation of actin assembly [43]. Thus, increased expression of both genes in plants correlates with the superior ability to tolerate flooding during emergence (Table 1; Figure 1). These findings indicate that there are two complex mechanisms of flooding tolerance during seedling emergence: (i) Increased mobilization of seed reserves and (ii) the ability to maintain root and shoot growth during seedling emergence under flood. 

### 3.3. Tolerance to Flooding during the Vegetative Stage

Flooding after the initial plant establishment elicits responses that differ across plant species and dependent on the speed and duration of flooding occurrence [5,23]. During initial growth, flooding stress-tolerant genotypes of rice can react by activating adaptation mechanisms to escape, or to withstand, flooding until the stress is alleviated by the formation of aerenchyma [44]. Although tolerance to flooding in rice is associated with aerenchyma, this mechanism of tolerance to flooding after establishment is also reported in other species, such as *Echinochloa spp.*, *Sagittaria montevidensis*; *S. trifolia*, *Cyperus spp.*, and *Rumex palustris* [5], or in flood-tolerant leguminous weeds of rice, such as *Sesbania herbacea* and *Aeschynomene virginica*. The presence of aerenchyma in flood-tolerant plants is also a snorkeling strategy. Aerenchyma is a plant tissue with a high proportion of intercellular spaces. The aerenchyma facilitates the diffusion of gases between the flooded and non-flooded organs of the plant, which allows the aerobic respiration of root cells [45]. The formation of aerenchyma in plants as a response to flooding is mediated by ethylene, inducing the activity of xyloglucan endotransglycosylase (*XETP*—Figure 1), which induces cell wall degradation. However, the formation of aerenchyma also requires reactive oxygen species (ROS) and Ca^2+^ in rice [46] and wheat [44]. In rice, ethylene or its precursor 1-aminocyclopropane-1-carboxylic acid (ACC) stimulates the formation of aerenchyma in roots [1]. 

The presence of aerenchyma alone does not guarantee the diffusion of O_2_ to the root tips, because of radial oxygen loss (ROL). The rate of long-range oxygen diffusion from aerial shoots to the submerged roots is maintained by a constitutive, or induced, an apoplastic barrier that prevents the ROL [1]. This barrier is typically composed of a suberized lamella that is formed in the exodermal/hypodermic space, near the tip of the root, and by lignified sclerenchyma/epidermal cells [23]. In this way, an efficient barrier is created that minimizes ROL and increases oxygen delivery to the root tip [44].

Genes encoding non-symbiotic hemoglobins (*NSHB1*) was observed in rice under flood conditions as a complementary mechanism for oxygen delivery to root tips, maintaining cell energy levels under conditions of hypoxia [47] (Figure 1). Hemoglobins react with nitric oxide (NO) produced under conditions of hypoxia, favoring an alternative route of respiration and transport of electrons in the mitochondria of plants under flood. Under hypoxia, NSHB1 reacts oxidatively with NO, producing nitrate and stimulating the consumption of NADH, thereby releasing NAD^+^ that can be reused in glycolysis [47]. These mechanisms, acting alone or concomitantly, certainly occur in flooding-tolerant weeds. However, the specific function in different species is not known.

Aerenchyma formation is crucial in cases of complete submersion of the plant, but aeration is more efficient when the aerial part can grow above the water level [48]. Some plant species growing in flooded environments have evolved the ability to elongate porous shoots under flooded conditions to facilitate the uptake of oxygen from the atmosphere and to transport it to the roots via the aerenchyma. This adaptation has a high energy cost and is restricted to native species in environments characterized by shallow, but prolonged flooding [24]. These flood-adapted plants possess the mechanism described as low O_2_ escape syndrome (LOES), which is observed in some rice varieties and *Rumex palustris* [22].

The submersion escape mechanism is activated by low oxygen concentration and promoted by ethylene, gibberellic acid, and auxin, which collectively promote underwater growth and stretching [21]. The modulation of the response to flooding in rice is a function of the *SNORKEL* (SK) locus, responsible for stem elongation under deep water [20] (Figure 1). This locus encodes two ethylene-inducible transcription factors, group VII (Ethylene Response Factor (ERF): *SK1* and *SK2* [23]. The expression of these genes promotes elongation of the rice internodes, enabling plants to elongate underwater at a rate of 25 cm d^−1^ in the flood-tolerant variety C9285 [20]. The signaling effects of *SK1* and *SK2* on the downstream events, which culminates in plant elongation is unclear, but act on GA biosynthesis [24]. The expression of the gene SK1 was associated with flooding tolerance in ITJ03 weedy rice genotype and was not observed in flood-sensitive rice cultivars IRGA 417 [13].

The escape response to flooding is beneficial if the plant reaches the surface of the water before the plant dies. This is a high-energy-cost strategy and could exhaust the plant’s carbon reserves before emergence from deep flood [22]. Thus, some species have evolved another mechanism of flood tolerance based on low O_2_ quiescence syndrome (LOQS), which was identified in local rice cultivars from Asia [21]. This tolerance mechanism also involves ethylene, with the *SUB1* locus encoding two to three ERF-VIIs, including *SUB1A*, which regulates metabolism and extends the rice survival under complete submersion to up to two weeks, as well as their recovery and regrowth after the stress ends [22]. Ethylene is accumulated up to saturation level in the plant under flooding, but the ABA and GA concentrations are constant. This tolerance mechanism is also present in the weed *Rumex acetosa,* exhibiting metabolic adjustments consistent with energy conservation [49]. In this way, the growth of the petiole and other energy-demanding processes are suppressed, allowing the conservation of resources until the flood recedes [24]. 

The SUB1A represses ethylene biosynthesis, thereby inhibiting ethylene-mediated elongation in rice [23] (Figure 1). SUB1A also enhances the transcription of Slender Rice1 and Slender RiceLike1, which negatively regulates GA responses and consequently diminishes plant growth [24]. The submergence of rice genotypes (Table 1) overexpressing *SUB1A* revealed that the reduction of growth is a result of the lower expression of genes associated with cell elongation, starch metabolism, and induction of fermentation genes [9,23]. The flood tolerance, affected by *SUB1A*, is accomplished by using carbohydrate reserves, preventing energy crisis during submersion [24]. Plants that have not exhausted their carbohydrate reserves at the end of flooding can resume regular development. However, when the flood subsides, the plants will be exposed to high light and oxygenation, resulting in ROS production and increased lipid peroxidation [50]. As a result of the damage to cell membranes, especially to the root, hydraulic conductance is minimized or halted, resulting in plant dehydration [50]. Therefore, the snorkeling strategy must involve more factors than just plant survival during flooding. It must also involve the ability to mitigate subsequent dehydration and oxidative stress, restore photosynthetic activity, and resume growth [24] post-flood or post-submersion. The mechanisms that allow species to tolerate or escape submersion and resume normal metabolism after flooding are not yet fully understood. The large diversity of wild *Oryza* species, weedy rice ecotypes, and other species thriving in flooded environments presents an opportunity to discover novel strategies to escape and survive periods of O_2_ deficiency. Although knowledge about flooding tolerance is still in its infancy, some of this information may be used to develop mitigation strategies for weedy rice and other weeds in flood-irrigated crops.

## 4. Biotechnological Approaches for Managing Flood-Tolerant Weeds

### 4.1. TAC-TIQUES Strategy

The *TAC-TIQUES* strategy involves using multiple copies of transposons mounted on a cassette under the control of an inducible promoter [14,51]. This technique was developed originally for the control of insects and was adapted for the mitigation of herbicide resistance in weeds. Many copies of the original transposons are engineered into the transgene cassette and transferred to the target species, such as rice. Introgression of these transposons occurs at multiple loci in different chromosomes, which results in trait expression across all offsprings [51]. In that sense, the crop would be transformed with a gene construct containing a herbicide-tolerant gene and a transposon containing a gene of susceptibility to another herbicide with a different mode of action. This approach was used successfully to obtain transgenic rice resistant to glyphosate introduced in tandem with an RNAi cassette suppressing the expression of the bentazon detoxifying enzyme CYP81A6, resulting in susceptibility to bentazon [52]. In this case, when gene flow occurs from the GM cultivated rice to weedy rice, the glyphosate-resistant hybrid weedy rice and volunteer GM rice plants can be selectively killed in the next crop by the application of bentazon. Thus, this strategy can be used to mitigate gene flow between a crop and its weedy relative.

The use of transposons containing modified KEV (*Kevorkian [chemically induced suicide]*) genes that promote susceptibility to flooding could reverse tolerance to flooding stress in weedy rice. One hypothetical application of this approach is the insertion (into rice) of a KEV containing two genes—the first, endowing resistance to a herbicide (or other desired trait) and the second, encoding RNAi that prevents mobilization of carbohydrates during germination in submerged condition. The putative gene to be silenced is *RAmy3D*. This gene is induced under submersion conditions, but is repressed to background levels under aerobic condition [42]. With this approach, it is expected that the germination of transformed, cultivated rice would occur regularly under aerobic condition. However, the outcrosses with weedy rice and the volunteer transformed rice would be controlled by flooding because of inability to germinate under anoxia. To achieve this, the transformed crop can be planted as pre-germinated seeds in shallow-flooded soil, allowing the establishment of the crop, but not the weedy rice outcrosses or volunteer GM rice. This approach will gradually eliminate weedy rice from the field, since gene flow from cultivated rice occurs at a rate of 0.03% or up to 1.26% with hybrid rice [53,54].

### 4.2. RNAi Strategy 

RNA interference (RNAi) is a sequence-specific, post-transcriptional gene silencing mechanism, induced by double-stranded RNA (dsRNA). This technology has been proposed for the mitigation of resistance in weeds and insects. The foliar application of specific dsRNA on tomato prevented the formation of actin in Colorado potato beetle, providing a new management tool for this insect [55]. This approach entails designing an RNAi that is based on the mRNA sequence of the target gene of the pest to be controlled [56]. The target gene needs to be identified and sequenced; complementary RNAi is then synthesized and incorporated into a silencing complex [57]. In doing so, the RNAi agent degrades or silences any transcript that shares the same sequence. A useful RNAi agent is one that does not cross-react with other genes, which could impair normal plant function. 

The BioDirect^TM^ technology, is an example of an application of gene silencing by RNAi for weed control [58]. The concept is to use dsRNA to silence the gene encoding the site of action of a herbicide in resistant weeds. For example, glyphosate-resistant weeds were sprayed with dsRNA to silence the allele associated with resistance to glyphosate [59]. This pertains to weeds that acquired resistance to glyphosate by producing multiple copies of the glyphosate target enzyme, EPSPS (*5-enolpyruvyl-shikimate-3-phosphate synthase*). Reducing the expression of EPSPS restores the susceptibility to glyphosate. A similar approach could be developed independent of herbicide resistance, and instead of targeting genes present only in the problem weed.

The knowledge of specific genes or alleles, such as those endowing flood tolerance, would allow the development of RNAi agents specific to problematic flood-tolerant weeds. This approach has potential for use in some flood-tolerant weed species, such as *Echinochloa spp*., *Sagittaria montevidensis*, *S. tripolla*, and sedges, where gene flow with rice cannot deliver the mitigation genes. In this way, the weed species would become sensitive to flooding while the crop remains normal (Figure 2). The control of major problematic, flood-tolerant weeds could then be attained with natural or artificial flooding. However, there is a major knowledge gap between the theoretical models and the practical applications of this technology mainly regarding efficiency in the inhibition of the target species, evolution of resistance, environmental stability, and specificity across non-target species. Further studies about flooding tolerance in weeds are needed to enrich our foundational knowledge for moving forward with the development of novel biotechnological weed mitigation methods.

## 5. Conclusions

There is a large diversity of strategies for flooding tolerance in rice across several stages of plant development. Weed adaptation to flooding is jeopardizing the utility of water as a primary non-chemical tool for weed control in flooded rice cropping systems. Flood-adapted weeds evolved one or more mechanisms of tolerance to flooding, which increase their invasiveness and competitiveness in cultivated areas. Information about flooding tolerance is available mainly in rice cultivars and a few other model crop species. Carbohydrate mobilization and maintenance of metabolism under anaerobic conditions are the main strategies of tolerance to flooding during germination, emergence, and initial plant establishment. These adaptive responses are mediated especially by the *RAmy3D*, *ADH*, *PDC, SUS*, and *SNORKEL* genes. In vegetative development, the presence of aerenchyma guarantees the aeration of submerged organs, as well as quiescence and escape strategies mediated by the *SUB1A* and *SNORKEL* genes, respectively, ensuring the perpetuation of weeds in flooded environments. Biotechnology may enable the development of novel tools for weed management with the use of transposable elements and RNAi, but knowledge about this technology is limited. Advances in genetic engineering and in the knowledge about flooding tolerance could revolutionize weed control mainly in the rice crop.

## Figures and Tables

**Figure 1 genes-11-00975-f001:**
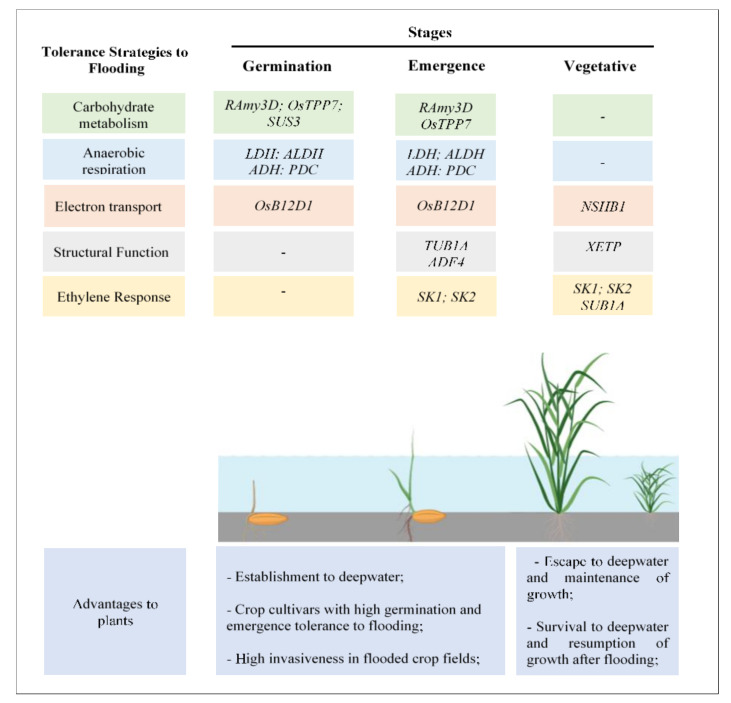
Scheme illustrating the main genes involved in tolerance to flooding at different plant development stages and potential advantages conferred to weeds. UFRGS, June 2020. RAmy3D (Rice α-amylases); SUS3 (sucrose synthase 3); OsTPP7 (trehalose-6-phosphate phosphatase); LDH (lactate dehydrogenase); ALDH (aldehyde dehydrogenase); ADH (alcohol dehydrogenase); PDC (pyruvate decarboxylase); OsB12D1 (named OsB12D1); NShB1 (non-symbiotic hemoglobins-B); TUB1A (tubulin a-1chain); ADF4 (actin factor 4); XETP (xyloglucan endotransglicosylase); SK1 (Snorkel1) SK2 (Snorkel2); SUB1A (*Submergence-*1).

**Figure 2 genes-11-00975-f002:**
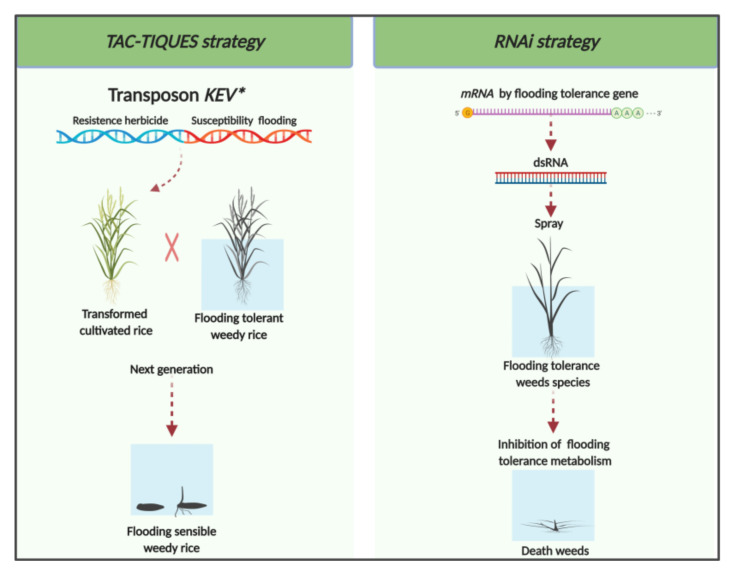
Scheme illustrating the potential application of biotechnological approaches for managing flood-tolerant weeds. UFRGS, June 2020.

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
