# Peer review of "Snorkeling Strategy: Tolerance to Flooding in Rice and Potential Application for Weed Management"

_genes, 2020, doi:10.3390/genes11090975_

Round 1
Reviewer 1 Report
Major concerns are;
In this review, utilization of gene silencing technology like RNAi or transposons is proposed to be potentially beneficial and effective measures to control rice weeds under flood in rice production. But unfortunately, this proposal seems not validated. The rationale states that horizontal transgene transmission between cultivated rice and paddy weeds is unlikely. However, in the first place, even if it might be possible to control certain gene-silenced paddy weeds under submergence, it will be difficult or impossible to control a variety of resistant weeds themselves growing in the actual paddy field ecosystem. This proposal of biotech-based control should be more thoroughly and critically examined.
Author Response
Reviewer 1
Major concerns are;
In this review, utilization of gene silencing technology like RNAi or transposons is proposed to be potentially beneficial and effective measures to control rice weeds under flood in rice production. But unfortunately, this proposal seems not validated. The rationale states that horizontal transgene transmission between cultivated rice and paddy weeds is unlikely. However, in the first place, even if it might be possible to control certain gene-silenced paddy weeds under submergence, it will be difficult or impossible to control a variety of resistant weeds themselves growing in the actual paddy field ecosystem. This proposal of biotech-based control should be more thoroughly and critically examined.
Regarding horizontal gene flow from rice to weeds - Except with its weedy relative (weedy rice), cultivated rice is not genetically compatible with other weeds in rice. Therefore, biotechnology-based weed management tools can be used generally in rice production with no risk of gene flow to unrelated weeds. The sprayable RNAi agent (against flooding tolerance) for weed control is topical and transient. This will not pose horizontal gene flow risk. The TAC-TIQUES strategy is designed to kill the weedy rice. Gene flow will not occur when the weedy rice is controlled. However, as with any tool, there is no guarantee for 100% control of the target 100% of the time. We have to make allowance for biological variability. Therefore, any of these tools will have to be used in conjunction with all other sound weed management practices for rice production. These biotechnological tools do not eliminate the need for supplemental herbicides nor negate best agronomic practices for rice production.
Response: Regarding the approach of using RNAi to control rice weeds in rice by silencing flooding tolerance genes – This, indeed, is in theory at this time. Gene silencing has not been tested yet in this context. But we are presenting this as one of the future trajectories of research and discovery for weed management in rice. The RNAi technique has been proven to work in other systems, i.e. with insects, nematodes, and even glyphosate-resistant Amaranthus. To date, it still needs tweaking in those systems. Nevertheless, we cannot avoid presenting the potential application of this technique in rice weed management. The research progress in weed genomics and advancements in biotechnological techniques will facilitate the discovery and development of novel molecular tools for weed management.
Additional information about that was added in the lines 74-87 of the current version. In addition, a Figura about that was added as suggested by the reviewer 2.
Reviewer 2 Report
This review entitled “Snorkelling Strategy: Tolerance to Flooding in Rice and Potential Application for Weed Management” is addressing a timely issue since weed management is predicted to become more important. The authors proposed the idea of mitigating flood resistance of weeds by combining the knowledge obtained in rice and biotechnology approach. Below I have a number of suggestions that the authors may consider in the further process of improving the manuscript.
Major point
This review describes mitigation of flooding tolerance using transposons and RNAi as new approaches to weed management. However, the figure in the manuscript only introduces the findings revealed in rice. Many reviews have already reported the flooding tolerance mechanism in rice. Therefore, it is easier for the reader to understand the main points of the manuscript by showing a schematic diagram of the chapter 4.
Minor points
- P2, lines 68-79.
These sentences are vague and feels difficult to understand for readers who are not familiar with this area.
- P2, lines 94-95, ‘SNORKEL (SK) locus…’.
This sentence is misleading, as if SK controls the production of ethylene, gibberellin and auxin. The authors should change these expressions if there is no concrete evidence that SK controls these phenomena.
- Table 1.
At first, I misunderstood that each tolerant genotype corresponds to each main gene in 'Germination and early growth' line. Probably it is not? Please improve the notation.
- P6, lines 233-235,’…, particularly in weeds, correlates with the superior ability…’
Is reference 42 described about rice? What is the basis for the sentence about weeds?
- P7, lines 280-281.
The authors should describe that SK regulates stem elongation, not cell elongation, because there is no data on cell elongation in ref 19.
Author Response
Reviewer 2
Comments and Suggestions for Authors
This review entitled “Snorkelling Strategy: Tolerance to Flooding in Rice and Potential Application for Weed Management” is addressing a timely issue since weed management is predicted to become more important. The authors proposed the idea of mitigating flood resistance of weeds by combining the knowledge obtained in rice and biotechnology approach. Below I have a number of suggestions that the authors may consider in the further process of improving the manuscript.
Major point
This review describes mitigation of flooding tolerance using transposons and RNAi as new approaches to weed management. However, the figure in the manuscript only introduces the findings revealed in rice. Many reviews have already reported the flooding tolerance mechanism in rice. Therefore, it is easier for the reader to understand the main points of the manuscript by showing a schematic diagram of the chapter 4.
Response: The figure is presented in attachment
Minor points
- P2, lines 68-79. These sentences are vague and feels difficult to understand for readers who are not familiar with this area.
Response: Details were added and are highlighted in red in the lines 69-88 of the current version.
- P2, lines 94-95, ‘SNORKEL (SK) locus…’. This sentence is misleading, as if SK controls the production of ethylene, gibberellin and auxin. The authors should change these expressions if there is no concrete evidence that SK controls these phenomena.
Response: A reference related to that was added (Hattoti, et al, 2009) supporting the information related with the SK locus
- Table 1. At first, I misunderstood that each tolerant genotype corresponds to each main gene in 'Germination and early growth' line. Probably it is not? Please improve the notation.
Response: The note “*One or more genes can be associated with the same tolerance mechanism” was added in the Table footnote.
- P6, lines 233-235,’…, particularly in weeds, correlates with the superior ability…’ Is reference 42 described about rice? What is the basis for the sentence about weeds?
Response: The description “particularly in weeds” was removed.
- P7, lines 280-281. The authors should describe that SK regulates stem elongation, not cell elongation, because there is no data on cell elongation in ref 19.
Response: The word “cell” was replaced by “stem”.

Reviewer 3 Report
This paper presents a good overview of various mechanism of how rice copes with flooding in different stages of development and what is the current status of knowledge about flooding mechanisms in weedy rice and other rice-related weeds and the potential technologies for weed management. The authors touched on known key genes in rice contributing tolerance to anaerobic germination and early growth and during vegetative stage and some recent findings of how those genes discovered in rice contribute to flooding tolerance in weedy rice. The authors then shared their future outlook on using RNAi and transposons as a potential new tool for weed management.
I do not see any major problems with this review, but I have some minor comments for further clarification, as follows:
- Figure 1: Please provide evidence or reference(s) that SK1 and SK2 are expressed during emergence.
- L190-192: Here the authors mentioned that OsB12D1 showed function in both anaerobic germination and emergence; however, it is not the case in Fig.1.
- Figure 1: It seems to me that OsTPP7 has function in both flooding during germination and emergence. Other than on coleoptile/young seedling, the gene is also strongly expressed in various parts of the seeds.
- Figure 1: The cartoon of seedling under vegetative stage only depicts the ‘escape strategy’ but not the ‘quiescence strategy’. It would be good to include both.
- Unfortunately, I cannot further verify the extent of gene expression study on the 158 weedy rice collection that the authors referred to (Kaspary et al.) since the paper is still in press. In this case, perhaps the authors can give a bit more details on the results of their study.
Author Response
Review 3
Comments and Suggestions for Authors
This paper presents a good overview of various mechanism of how rice copes with flooding in different stages of development and what is the current status of knowledge about flooding mechanisms in weedy rice and other rice-related weeds and the potential technologies for weed management. The authors touched on known key genes in rice contributing tolerance to anaerobic germination and early growth and during vegetative stage and some recent findings of how those genes discovered in rice contribute to flooding tolerance in weedy rice. The authors then shared their future outlook on using RNAi and transposons as a potential new tool for weed management.
I do not see any major problems with this review, but I have some minor comments for further clarification, as follows:
- Figure 1: Please provide evidence or reference(s) that SK1 and SK2 are expressed during emergence.
Response: The study developed by Kaspary et al (Accepted in Weed Research journal) presents evidence about the expression of SK during emergence. The Abstract of the article is presented below.
- L190-192: Here the authors mentioned that OsB12D1 showed function in both anaerobic germination and emergence; however, it is not the case in Fig.1.
Response: Suggestion accepted, and the information was added in the column of emergence. It was already present in the column of germination.
- Figure 1: It seems to me that OsTPP7 has function in both flooding during germination and emergence. Other than on coleoptile/young seedling, the gene is also strongly expressed in various parts of the seeds.
Response: The information was added in the column Germination, and it was already described in the column Emergence.
- Figure 1: The cartoon of seedling under vegetative stage only depicts the ‘escape strategy’ but not the ‘quiescence strategy’. It would be good to include both.
Response: A new plant was added indication now both strategies.
- Unfortunately, I cannot further verify the extent of gene expression study on the 158 weedy rice collection that the authors referred to (Kaspary et al.) since the paper is still in press. In this case, perhaps the authors can give a bit more details on the results of their study.
Response: The abstract of the accepted article is described below.
Genes related to flooding tolerance during germination and early growth of weedy rice. Evolution of flooding tolerance in weedy rice had occurred in several rice growing regions, but the genes related with this process and the environmental effect are unknown. The objective of this study was to analyze the expression of genes related to flooding tolerance in response to temperature and flooding during the initial establishment of weedy rice. The experiments were carried out with the rice cultivars IRGA 417 and Nipponbare, which are sensitive to flooding, and the weedy rice ITJ03 and AV04 genotypes, which presented high and intermediate tolerance to flooding, respectively. Expression of genes related to reserve mobilization, anaerobic respiration, escape, and quiescence strategies were analyzed at periods up to 24 days after sowing (DAS). The flooding tolerance of weedy rice genotype ITJ03 is associated with the expression of RAmy3D and OsTPP7 genes, which are involved with the mobilization of carbohydrate reserves, ADH1 and ADH2 which participate in anaerobic respiration, and culminates with the rapid elongation of the coleoptile and emergence provided by the SNRKL1 gene. The expression of the genes PDC1, SUS3 and SUB1 was not directly related to flooding tolerance in weedy rice, as compared to the importance of expression of these genes in cultivated rice. A temperature of 20°C reduced the expression of the RAmy3D, ADH2 and SNRKL1 genes. Low temperature had a negative effect on the establishment of weedy rice, and the joint use of these factors may be a viable strategy for improving the control of this weed in rice paddies.